# Effects of Organophosphorus Flame Retardants on the Dissipation Factor of Flame-Retardant Polymers

**DOI:** 10.3390/polym17091254

**Published:** 2025-05-05

**Authors:** Peng Jin, Qiang Yao, Weihong Cao, Jinhao Sun, Yueying Zhao

**Affiliations:** 1School of Chemistry and Chemical Engineering, Jiangxi University of Science and Technology, Ganzhou 341000, China; jinpeng@nimte.ac.cn; 2Key Laboratory of Bio-Based Polymeric Materials Technology and Application of Zhejiang Province, Ningbo Institute of Materials Technology and Engineering, Chinese Academy of Sciences, Ningbo 315201, China; caoweihong@nimte.ac.cn (W.C.); sunjinhao@nimte.ac.cn (J.S.); zhaoyueying@nimte.ac.cn (Y.Z.); 3Ningbo College of Materials Engineering, University of Chinese Academy of Sciences, Ningbo 315201, China

**Keywords:** dissipation factor, conformation, crystallizability, PMMA, PS, PPO, flame retardant

## Abstract

To understand the effect of the hydroxyl group and processing temperatures on dielectric losses of flame retardants and flame-retardant polymers, the performance difference between 6-methyldibenzo[c,e][1,2]oxaphosphinine 6-oxide (DOPO-Me) and 6-(hydroxymethyl)dibenzo[c,e][1,2]oxaphosphinine 6-oxide (DOPO-HM) has been investigated, respectively, in non-polar and polar polymers at 7–20 GHz. DOPO-HM and DOPO-Me differ by only one OH group. The former demonstrates a lower dissipation factor (Df) than the latter, owing to hydrogen bonds. In polystyrene and crosslinked polyphenylene oxide, both flame retardants increase a dielectric loss of flame-retardant polymers, with DOPO-HM being less detrimental because of its higher crystallizability and lower plasticization. In polar poly(methyl methacrylate) (PMMA), conformational changes in PMMA main chains caused by flame retardants and high processing temperatures lead to an early Df drop of PMMA at low loadings of the flame retardants. At high loadings, a change in the physical form of flame retardants from a primitive crystalline state to an amorphous state increases a dielectric loss of flame retardant PMMA, with DOPO-HM resulting in a slightly higher dielectric loss than DOPO-Me. These results prove that the effect of a hydroxyl group in organophosphorus structures on the dielectric loss of flame-retardant polymers is crucially dependent on its interaction with the polymer matrix.

## 1. Introduction

Modern wireless technology features high peak data speeds and ultra-low latency. Since data rates and signal loss are largely decided by the dielectric properties of materials, several polymers with low dielectric constant (Dk) and dissipation factor (Df) have been developed and used as a base material for printed circuit boards (PCBs). For example, thermosetting polyphenylene oxide (PPO), cyanate esters, polyimide, polyolefins and polytetrafluoroethylene have been employed in high-frequency microelectronics with low signal loss [1,2,3,4,5,6,7,8].

However, some of the above polymeric materials do not have sufficient flame retardancy. To pass the mandatory flammability tests such as a UL-94 standard [9], a significant amount of flame retardants has to be added. Consequently, flame retardants exert serious influence on the PCB dielectric properties. Additionally, the current trend toward non-halogenated flame retardants has channeled the electronics industry to use organophosphorus flame retardants instead of brominated flame retardants. As a result of this impetus, some promising organophosphorus flame retardants with stable P-C bonds have been developed, for example, derivatives of 6H-dibenzo[c,e][1,2]oxaphosphinine 6-oxide and diphenylphosphine oxide [10,11,12,13,14].

These organophosphorus flame retardants perform well in terms of flame retardancy; however, they are quite expensive, which poses a great challenge for the industry in popularizing halogen-free products. In order to develop low-cost yet high-performance organophosphorus flame retardants for modern communication applications, a relationship between the structures of organophosphorus flame retardants and the dielectric properties of flame-retardant polymers needs to be firstly established. Currently, this relationship has not been clear or at times is misunderstood. In particular, some patents claim that the presence of a hydroxyl group, one of the most polar groups, in structures of organophosphorus flame retardants is disadvantageous for high-frequency PCB design because of its easy polarization, which foreseeably causes an adverse effect on dissipation factor [15,16,17]. Consequently, organophosphorus compounds with an OH group in their structures have been largely ignored as flame retardant options in high-frequency and high-speed PCBs so far.

Nevertheless, considering beneficial effects brought by a hydroxyl group on the melting points and crystallinity of small molecules, which can counter the easy polarization of the hydroxyl group, the influence of an OH group in organophosphorus flame retardants on dielectric losses simply cannot be inferred. This is rather different from the dielectric constant, which has been satisfactorily predicated by various models [18]. In addition, organophosphorus compounds with a P–CH_2_OH moiety can be readily accessed via an economic Pudovik reaction [19]. Thus, organophosphorus compounds with an OH group deserve to be explored and the effect of hydroxyl groups needs to be elucidated.

In this work, we study the effect of a hydroxyl group on the dielectric losses of flame retardants and flame-retardant polymers. By comparing Df values of 6-methyldibenzo[c,e][1,2]oxaphosphinine 6-oxide (DOPO-Me) with 6-(hydroxymethyl)dibenzo[c,e][1,2]oxaphosphinine 6-oxide (DOPO-HM) and their blends with polymers ranging from non-polar to polar, we have demonstrated that an OH group can have a positive impact on the dielectric loss of flame retardants and that it is the interaction between flame retardants and polymeric materials that is a key to deciding the Df values of flame-retardant polymers. Our study will help open up a new avenue in the development of high-performance, yet affordable, flame retardants and assist product designers to formulate and choose right flame retardants for their PCBs.

## 2. Experimental Section

### 2.1. Materials

Polystyrene (PS, 143E) was purchased from SINOPEC YANGZI PETROCHEMICAL Co., Ltd. (Nanjing, China). Poly(methyl methacrylate) (PMMA, CM-205) was acquired from Zhenjiang Chimei Chemical Co., Ltd. (Zhenjiang, China). Methacrylate end-capped polyphenylene oxide (PPO, SA9000) was obtained from Saudi Basic Industry Corporation (Riyadh, Saudi Arabia). 9,10-dihydro-9-oxa-10-phosphaphenanthrene-10-oxide was purchased from Shanghai Maclin Biochemical Technology Co., Ltd. (Shanghai, China). Divinylbenzene (DVB), diisopropylbenzene peroxide (DCP), toluene and paraformaldehyde were purchased from Shanghai Aladdin Biochemical Technology Co., Ltd. (Shanghai, China). All materials were used without any further purification.

### 2.2. Synthesis of Flame Retardants

DOPO-Me and DOPO-HM are known compounds (see their structures below, Figure 1). They were synthesized according to [20,21].

### 2.3. Preparation of Flame-Retardant Polymers

**Flame retardant:** powder DOPO-Me and powder DOPO-HM were each pressed into a 2.0 mm thick plate in a vulcanizing press at 60 °C under 9.8 Mpa for 5 min.

**Flame retardant PS or PMMA:** PS, PMMA, DOPO-Me and DOPO-HM were dried in an oven at 80 °C for 4 h before use. All PS or PMMA samples containing a pre-determined weight of DOPO-Me or DOPO-HM were prepared in a Brabender mixer with a rotor speed of 50 rpm. The mixing time of each blend was five minutes. After mixing, samples were pressed into a film in a mold with a thickness of 0.60 mm by a vulcanizing press at a preset temperature. The plate pressure was 9.8 MPa and the pressing time was 5 min followed by cooling. In a slow cooling procedure, the film was taken out of the press but kept in the mold and cooled by air while it was cooled in the press under pressure by circulating cold water in a fast cooling method. The preparation conditions for each sample, with a designated name, are listed in Table 1.

**Flame-retardant PPO:** PPO, DOPO-Me or DOPO-HM, DVB and 40 g toluene were mixed according to the formula listed in Table 2 at 70 °C for 10 min. Then, DCP was added and stirred for another 30 min at 70 °C. The solution or suspension so obtained was dried in a vacuum oven at 80 °C for 4 h. The chunk solid was subsequently cured on a vulcanizing press according to the procedures of 140 °C/4 h + 170 °C/1 h + 190 °C/1 h. The pressure of the press was 15.0 MPa. After curing, it was naturally cooled to 65 °C in the press under a pressure of 15.0 MPa.

### 2.4. Characterizations

^1^H NMR and ^31^P NMR of flame retardants or PMMA were performed in a Bruker AVANCE III 400 MHz nuclear magnetic resonance spectrometer (Bruker, Billerica, MA, USA) at 400 MHz and 162 MHz, respectively. Deuterated chloroform (CDCl_3_) was used as a solvent.

The melting points, crystallinity and glass transition temperatures (*T_g_*) of each flame retardant or flame-retardant polymer were obtained in a Mettler Toledo DSC1 differential scanning calorimetry (Mettler Toledo, Greifensee, Switzerland). Samples were heated from 30 °C to 250 °C at a heating rate of 10 °C/min under nitrogen. The weight of each sample was 5 to 8 mg. *T_g_* was taken from the first heating curve and was the average of three measurements.

The density of each flame retardant was determined in a density balance instrument (Mettler Toledo, Switzerland).

Dielectric properties of flame retardant and flame-retardant polymers were measured in a ZNA67 high-frequency vector network analyzer (Rohde & Schwarz, Munich, Germany) at 25 °C. Df and Dk values were the average of five measurements.

The solid-state NMR experiments were performed on an Agilent DD2 spectrometer operating at 150.78 MHz for ^13^C. A double resonance 4 mm T-3 MAS probe was used for all the experiments. The ^13^C spin-lattice relaxation time (T_1_) was acquired using a tancpxt1 sequence, with a contact time of 1 ms and a recycle interval of 2 s. The spinning rate was set to 8 kHz. The ^13^C chemical shifts were calibrated using adamantane (δ = 38.56 ppm).

## 3. Results and Discussion

### 3.1. Effect of OH on Df and Dk of Flame Retardants

The Df values of two flame retardants are plotted against the frequency in Figure 2. Clearly, except at a resonant frequency around 11 GHz, DOPO-HM has much lower dielectric losses than DOPO-Me in the measured range. This is rather interesting because the hydroxyl group in DOPO-HM actually produces the sought-after effect of lowering the Df values. Perceptibly, it is attributed to hydrogen bonds formed between P=O and OH, which slow down a dipole reorientation of the polar groups in DOPO-HM under a fast alternating electric field [22]. As a result, the dielectric loss of DOPO-HM is decreased. This effect is similar to that of hydroxyl group on the dielectric loss of epoxy or hydroxylated polypropylene, where hydrogen bonds contribute to a reduction in the dielectric loss, although hydrogen bonds are produced by an extraneous additive in the latter cases [23,24].

In addition, DOPO-HM produces smaller Dk values than DOPO-Me, which is also surprising. Low-Dk flame retardants are desirable in high-speed PCBs. On the basis of a density of 1.3855 g/cm^3^, which is higher than 1.3248 g/cm^3^ of DOPO-Me, DOPO-HM should have larger Dk values than DOPO-Me, since Dk is dependent on free volumes of molecules. An increase in free volumes decreases the dielectric constant of a molecule, owing to a reduced number of polarizable groups per unit volume [25]. Judging by its structural similarity but lower density, DOPO-Me presumably has greater free volumes than DOPO-HM. However, hydrogen bonds can cause a significant drop in permittivity [18] and this factor obviously exceeds the influence of free volumes. Therefore, besides a lower Df value, DOPO-HM also has a lower Dk value than DOPO-Me.

### 3.2. Effect of OH on Df of PS/FR

Polystyrene is a non-polar amorphous polymer and was chosen as a model polymer for hydrocarbon polymers like styrene–butadiene–styrene copolymer and polybutadiene, which have been commonly used in PPO-based formulations. Considering that processing temperatures might affect the mixing of the flame retardant and polymer, two temperatures were used, i.e., flame-retardant polystyrene was processed at 150 °C and 190 °C, respectively, and obtained by cooling under pressure from the processing temperature. Since polystyrene has a resonant frequency around 8.6 GHz, the Df values of polystyrene/flame retardant (PS/FR) are not useful at this point. However, apart from this resonant frequency, the effects of flame retardants on the dielectric loss of flame-retardant polystyrene are clear in the range of frequencies from 7 to 20 GHz.

For PS-150-DOPO-Me, its Df values increase linearly with the amount of the flame retardant until they reach an inflection point around 5% of DOPO-Me, after which the Df values still remain a good linear relationship with DOPO-Me but the lines become less steep, as seen in Figure 3. The bending point at 5% DOPO-Me is apparently caused by a change in the neighboring environment of the flame retardant in PS. This explanation is readily supported by DSC results.

DSC thermograms indicate that DOPO-Me exists as an amorphous form in PS-150-DOPO-Me, because there is no melting peak observed for DOPO-Me. At low concentrations, each DOPO-Me molecule is surrounded by non-polar polystyrene chains. However, at high loadings, molecules of DOPO-Me start to aggregate. In line with this aggregation, the *T_g_* curve of PS-150-DOPO-Me also bends at 5% due to less plasticization per DOPO-Me molecule. The aggregation creates local potential gradients and increases the activation energy to rotate a polarized DOPO-Me molecule. Subsequently, the ability of P=O dipole realignment is reduced in an applied electric field and the rate of Df increases slows down at high loadings of DOPO-Me. However, the overall Df values continue to rise due to a high dielectric loss of amorphous DOPO-Me.

On the other hand, PS-150-DOPO-HM produces a somewhat different pattern. Although the Df values of PS-150-DOPO-HM increase linearly with the flame retardant content, the linear relationship is only found at very low loadings of DOPO-HM (0-0.5%), far narrower than the broad range (0-5%) found in PS-150-DOPO-Me, implying that DOPO-HM is easier to aggregate. After 1%, the Df values of PS-150-DOPO-HM reach a plateau, resulting from a balance of the opposite actions of an increasing loading of the flame retardant and a decreasing concentration of amorphous DOPO-HM. Indeed, DSC thermograms of PS-150-DOPO-HM show increasing crystallinity of DOPO-HM from 19% in 2.5%-DOPO-HM to 90% in 30%-DOPO-HM.

Compared with PS-150-DOPO-Me, PS-150-DOPO-HM has notably lower Df values at high loadings of the flame retardant. Even when cooled from a higher processing temperature which enables complete melting of DOPO-HM, PS-190-DOPO-HM still produces lower Df values at 20 GHz than PS-190-DOPO-Me, which has almost identical Df values as PS-150-DOPO-Me. Obviously, poorer plasticization of amorphous DOPO-HM, as evidenced by less reduced *T_g_* values shown in Figure 3, could not enhance mobility of PS chains as great as amorphous DOPO-Me does. In other words, DOPO-HM is more associated at high loadings and hence the Df increases slow down. From the above results, it can be seen that organophosphorus flame retardants with a hydroxyl group can actually be advantageous in non-polar polymers in keeping a low dielectric loss of the latter.

In addition, to quantify the Df values of amorphous DOPO-Me and DOPO-HM, linear regression of the Df values of the flame-retardant polystyrene as a function of the flame retardant content was used. The concentration of DOPO-HM was limited to a range from 2.5% to 10% due to a mass exodus beyond 10%. This would lead to a higher Df value of amorphous DOPO-HM because of fewer associations. However, it is still only slightly higher than the Df value of amorphous DOPO-Me, as seen in Table 3.

### 3.3. Effect of OH on Df of PMMA/FR

PMMA was used as a model polymer to study the effect of flame retardants on dielectric losses of polar polymers such as methacrylate end-capped PPO. ^1^H NMR reveals that the PMMA used in this study was atactic with syndio-triads of 51%, a significant fraction [26]. Since both the flame retardants and PMMA are polar, the solubility of DOPO-HM and DOPO-Me in PMMA is expected to be high. Indeed, all DOPO-Me and DOPO-HM dissolve in PMMA when their contents are equal to or less than 20%, as evidenced by the absence of peak of the flame retardants in X-ray diffraction of the flame retardant blends.

In terms of the dielectric loss of the flame retardant PMMA (PMMA/FR), except at two resonant frequencies of PMMA at 8.6 GHz and 13.7 GHz, their Df values generally increase with the flame retardant content at high loadings as shown in Figure 4 and Table 4. However, it is surprising to note that both DOPO-Me and DOPO-HM cause an initial Df drop of PMMA upon the addition of a very small amount of the flame retardants, albeit with DOPO-HM having a much smaller effect. The early Df reductions are rather unexpected because the presence of amorphous flame retardants should have led to an increase in the Df values at low loadings as it does at high loadings. The opposite results suggest that the flame retardants at low loadings may have decreased net polarity of PMMA, resulting in a drop in the dielectric loss. Since the dielectric loss of pure PMMA at high frequencies primarily arises from its ester group with other accompanied methylene group [27,28], the presence of the flame retardants might have changed the spatial arrangement of the polar ester groups which are dependent on specific conformations.

To test the above hypothesis, experiments were first carried out on PMMA with different cooling rates. It was reasoned that conformations of PMMA chains can be better preserved at a fast cooling rate than at a slow cooling rate so a different degree of certain spatial arrangement of the polar ester groups can be produced. Since PMMA main chains have been documented to exist dominantly in a trans-trans (tt) conformation in melts [29,30] and since extra extended chains, which has a tt conformation, are generated by high pressure used in fabricating test samples, there should be an above-normal concentration of tt conformations in PMMA at room temperature after rapid cooling. The tt conformation leads to a longer distance between intra-chain ester groups than trans-gauche (tg) or gauche-gauche (gg) conformations, as shown in Figure 5; as a result, the dipole–dipole interactions of intra-chain ester groups are weak and the Df values should increase.

Indeed, the PMMA sample obtained at a fast cooling rate (PMMA-170-quick) features Df values higher than that attained at a slow cooling rate (PMMA-170-slow), as noted in Table 4, fully in agreement with the above assumption. Moreover, when PMMA was rapidly cooled from 190 °C, PMMA-190 gained a higher Df value than PMMA-170. Apparently, the high pressure used in making PMMA samples produced more tt conformations at 190 °C than at 170 °C because of the higher mobility of PMMA chains at a higher temperature. Thus, there were even more tt conformations in PMMA-190 than in PMMA-170, which is strongly supported by an increased *T_g_* of the former, as shown in Table 4 [31].

Furthermore, solid-state NMR experiments were performed to extract the T_1_ relaxation time constant of carbonyl carbon of PMMA to distinguish dipole–dipole interactions in PMMA samples made at different processing temperatures. It was expected that dipole–dipole interactions, foreseeably originating from tg or gg conformations that lead to the curl-up of PMMA chains, would reduce the T_1_ value. In fact, PMMA-170-slow has a lower T_1_ value of 24.6 s than PMMA-170-quick, which itself has a lower T_1_ value of 27.0 s compared to that of PMMA-190-quick, 29.8 s, as listed in Table 4. Therefore, the T_1_ data clearly differentiate the dipole–dipole interactions in these PMMA samples and imply a relationship between the dielectric loss and tt conformations of PMMA, which is forged through dipole–dipole interactions.

Now, for PMMA/FR, particularly PMMA-170-DOPO-Me-quick, flame retardants work in another way, but with a similar effect, to change the distribution of tt conformations of PMMA. As inferred in Table 4, DOPO-Me acts as a plasticizer to effectively lower the glass transition temperature of PMMA. The plasticizer enables PMMA main chains to gain enough mobility [32] to overcome their torsional barrier and acquire tg or gg conformation, achieving a comparable effect as in slow cooling. As a result of forming the tg or gg conformations, which can facilitate dipole–dipole interactions of intra-chain ester groups, both the T_1_ and Df values reduce initially, as demonstrated in Figure 4 and Table 4.

However, intramolecular dipole–dipole coupling of ester groups can be easily destroyed by excess DOPO-Me by solvation. After 1%, the T_1_ values of PMMA-170-DOPO-Me-quick rise again and eventually reach a number around 28 s. On the other hand, the Df values of PMMA-170-DOPO-Me-quick continuously keep increasing, mainly due to a large Df value of amorphous DOPO-Me. In short, the initial decrease in the dielectric loss of PMMA/DOPO-Me at low loadings of the flame retardant is owed to an abnormally high concentration of tt conformation of PMMA chains, while its increase at high loadings can be largely attributed to the amorphous form of DOPO-Me.

On the other hand, DOPO-HM displays a much more complicated effect than DOPO-Me. At low loadings, DOPO-HM actually works as an anti-plasticizer to restrict the mobility of PMMA chains, as evidenced by the increased *T_g_* values of PMMA-170-DOPO-HM-quick (to see Table 4). Obviously, the anti-plasticization originates from hydrogen bonds between PMMA ester groups and hydroxyl groups of DOPO-HM [33,34]. Thus, more PMMA chains stay in extended conformation when they are rapidly cooled from a high temperature, which is consistent with an increased T_1_ value of PMMA-170-DOPO-HM-quick at 0.25% DOPO-HM. However, the values of the dielectric loss slightly decrease instead of increasing, due to hydrogen bonds that slow down dipole alignment of both ester groups and hydroxyl groups in an applied electric field [35]. So, the reasons for the changes in T_1_ and Df values of PMMA-170-DOPO-HM-quick are entirely different from those in PMMA-170-DOPO-Me-quick at low loadings.

Nevertheless, when PMMA-190-DOPO-HM-quick was cooled from 190 °C, its T_1_ values first decrease before returning to a normal value at 1% DOPO-HM. This pattern is closely analogous to the T_1_ of PMMA-170-DOPO-Me, implying that DOPO-HM now acts as a plasticizer rather than an anti-plasticizer when cooled from a higher processing temperature. Obviously, DOPO-HM works to help PMMA-190 main chains, which has an unusually high concentration of tt conformations, as suggested by its high *T_g_*, adopt tg or gg conformations and return to a normal distribution of each conformation. Accordingly, the levels of T_1_ as well as Df reduce for PMMA-190-DOPO-HM-quick at low loadings of DOPO-HM.

At high loadings, both Df values of PMMA-190-DOPO-HM-quick and those PMMA-170-DOPO-HM-quick increase and eventually become comparable, implying that a high loading of DOPO-HM can eliminate the influence of processing temperature. In any case, PMMA/DOPO-HM only experiences a moderately higher dielectric loss than PMMA/DOPO-Me at a high content of flame retardant.

Additionally, it is noted that when T_1_ of PMMA-190-DOPO-HM-quick returns to a normal value at 1% of the flame retardant, its dielectric loss still continuously decreases. Clearly, intramolecular dipole–dipole coupling of ester groups disappear at this loading level but there are still intermolecular hydrogen bonds that affect dipole alignment of both ester groups and hydroxyl groups in an applied electric field. Therefore, the Df value at 1% of DOPO-HM further reduces.

Similarly to the treatment in PS/FR, the Df values of the amorphous flame retardants were also extrapolated from the linear regression of Df vs. the flame retardant content. Good linear relationship is found at high loadings for both flame retardant blends obtained by cooling down from 170 °C. The results are listed in Table 3. It can be seen that the Df value at 20 GHz of amorphous DOPO-Me is very close to that obtained in PS/DOPO-Me. It also confirms that amorphous DOPO-HM has only slightly higher dielectric loss than amorphous DOPO-Me at the frequency range of 7–20 GHz. This suggests that organophosphorus compounds with a hydroxyl group can be good candidates for high-performance flame retardants, even in a polar polymer.

### 3.4. Effect of OH on Dfs of PPO/FR

Upon the addition of either flame retardant at 10%, the Df values of PPO/FR increase and are comparable at the same frequency, regardless of the structure of the flame retardant, because of enhanced mobility of crosslinked PPO chains and amorphous flame retardants, as shown in Figure 6. However, with a further loading, the Df values of PPO-DOPO-Me keep shifting upward, while those of PPO-DOPO-HM start to go downwards. Thus, PPO-DOPO-HM has lower Df values than PPO-DOPO-Me at high loadings. This trend is similar to that found in PS/FR and is clearly attributed to crystallizability of the flame retardants, which relies on their solubility. The solubility of DOPO-HM is lower than that of DOPO-Me in PPO. As a matter of fact, DOPO-HM always partially crystallizes in PPO, while DOPO-Me is completely soluble even at 30%. Therefore, although amorphous DOPO-HM has higher dielectric loss than amorphous DOPO-Me, polymer blends containing DOPO-HM do not necessarily possess higher Df values, because DOPO-HM tends to crystallize.

In addition, the theoretical Df values of PPO/FR were calculated based on the weight addition and crystallinity of each component. The Df values of amorphous DOPO-Me and DOPO-HM obtained in PMMA/FR were used for amorphous flame retardants. Figure 7 shows the comparison of theoretical curves vs. experimental curves. It can be seen that a generally fair-to-good agreement with measurements is obtained, especially at high loadings of flame retardants.

## 4. Conclusions

To develop high-performance organophosphorus flame retardants for modern communication applications and understand the structural effect, in particular the effect of a hydroxyl group, on the dielectric losses of flame retardants and flame-retardant polymers, the performance difference between DOPO-HM and DOPO-Me has been independently investigated in non-polar and polar polymers. DOPO-HM differs from DOPO-Me only by the addition of a hydroxyl group.

It has been demonstrated that DOPO-HM possesses lower Dk and Df values than DOPO-Me at 7–20 GHz because the former is able to form hydrogen bonds that slow down dipole reorientation of the polar groups. Both flame retardants increase the dielectric loss for non-polar polystyrene. However, DOPO-HM causes a slower Df increase than DOPO-Me due to higher crystallizability and poorer plasticization of DOPO-HM.

In polar PMMA, factors affecting the distribution of conformations of PMMA contribute to changes in Df values. At low loadings and a low processing temperature, both DOPO-Me and DOPO-HM cause an initial Df drop of PMMA. However, the reasons are different. For DOPO-Me, it acts as a plasticizer to reduce the *T_g_* of PMMA and facilitate the return of a tt conformation to a tg or gg conformation and hence both the T_1_ values of carbonyl carbon and the Df values of PMMA/DOPO-Me reduce initially. On the other hand, DOPO-HM initially works as an anti-plasticizer for PMMA-170; as a result, T_1_ increases, owing to fewer dipole–dipole interactions, while the Df value reduces, presumably because of hydrogen bonds between DOPO-HM and PMMA. Nonetheless, like DOPO-Me, DOPO-HM can also act as a plasticizer for PMMA-190, which was processed at a higher temperature. This is because there is a higher concentration of tt conformations in PMMA-190. In any case, PMMA/DOPO-HM only has moderately higher Df values compared to PMMA/DOPO-Me at the same high loadings at the measured frequencies.

Lastly, in a crosslinked PPO system, DOPO-HM performs better than DOPO-Me, as they do in PS. By utilizing the crystallinity of each component and based on the weight addition, the theoretical Df values of PPO/FR have been calculated. A generally fair-to-good agreement with measurements is obtained, especially at high loadings of flame retardants. The above results prove that the interaction between flame retardants and polymeric materials is really the key to deciding the effect of a hydroxyl group on the Df values of flame-retardant polymers. Consequently, organophosphorus structures with a hydroxyl group deserved to be explored in the development of high-performance, non-halogenated flame retardants for high-speed, high-frequency PCBs.

## Figures and Tables

**Figure 1 polymers-17-01254-f001:**
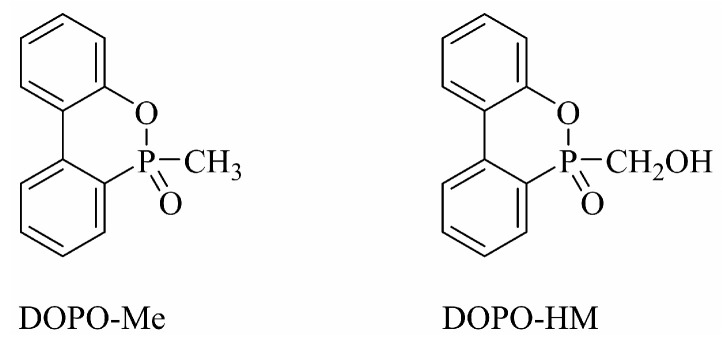
The structures of DOPO-Me and DOPO-HM.

**Figure 2 polymers-17-01254-f002:**
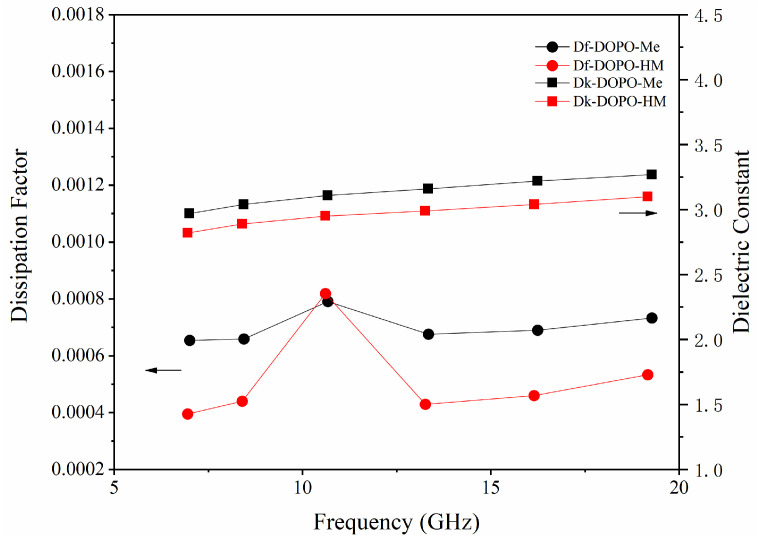
Df and Dk of flame retardants.

**Figure 3 polymers-17-01254-f003:**
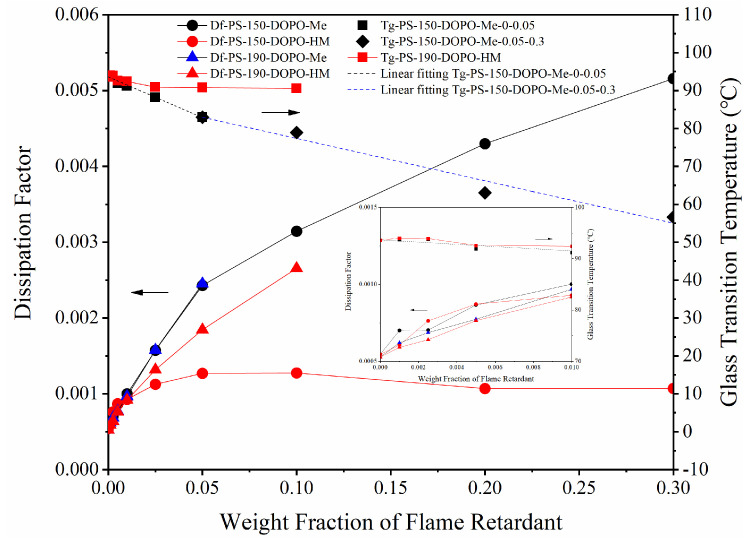
Df and *T_g_* of PS/FR (Df values are taken at 20 GHz).

**Figure 4 polymers-17-01254-f004:**
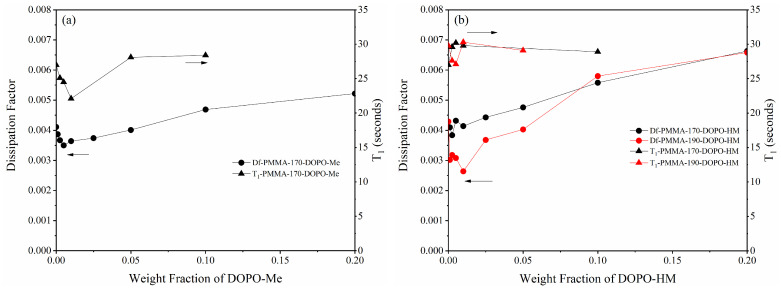
Df and T_1_ relaxation time constant of carbonyl carbon of PMMA/FR (Df values are taken at 20 GHz): (**a**) PMMA-DOPO-Me; (**b**) PMMA-DOPO-HM.

**Figure 5 polymers-17-01254-f005:**
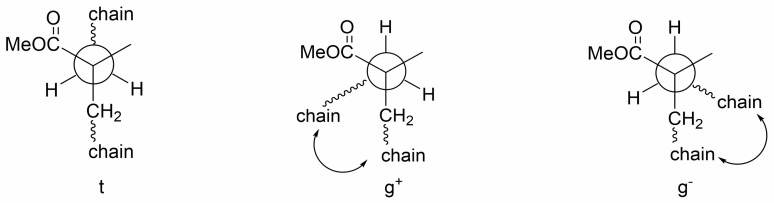
Newman projections of backbone t, g^+^ and g^−^ states of PMMA. g conformation leads to a curl-up of main chains, resulting in a shorter distance between intra-chain ester groups and hence more dipole–dipole interactions (ester groups in main chains not shown for simplicity).

**Figure 6 polymers-17-01254-f006:**
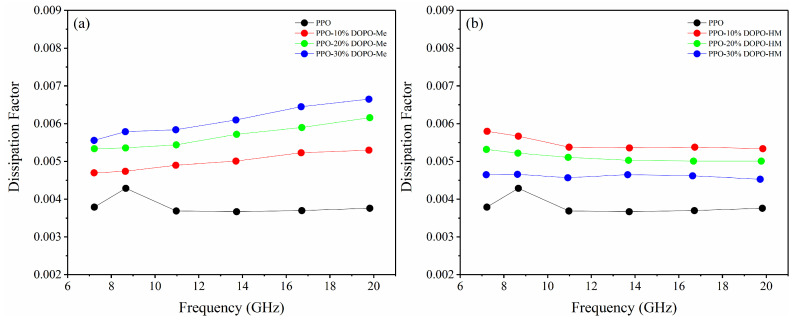
Df of PPO/FR: (**a**) PPO-DOPO-Me; (**b**) PPO-DOPO-HM.

**Figure 7 polymers-17-01254-f007:**
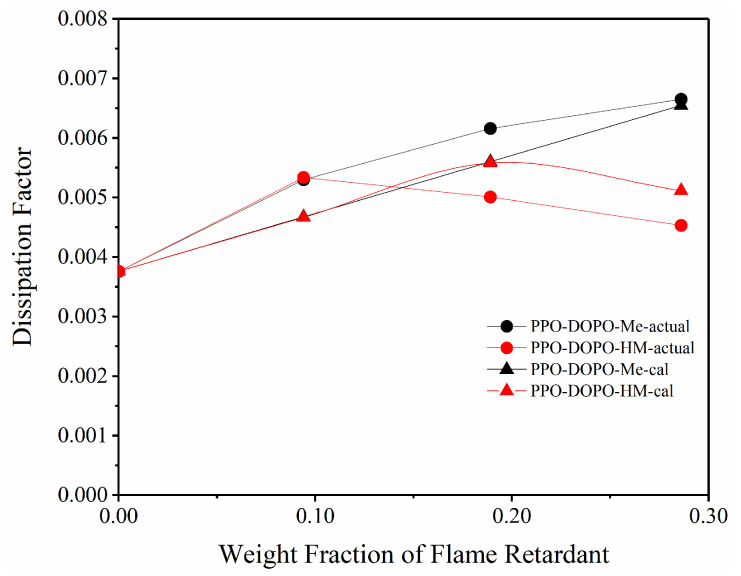
Predicted Df vs. actual values of PPO/FR at 20 GHz.

**Table 1 polymers-17-01254-t001:** Preparation conditions for samples.

Sample	Mixing Temperature (°C)	Pressing Temperature (°C)	Fast Cooling	Slow Cooling
PS-150-DOPO-Me	150	150	Y	
PS-150-DOPO-HM	150	150	Y	
PS-190-DOPO-Me	190	190	Y	
PS-190-DOPO-HM	190	190	Y	
PMMA-170-slow	160	170		Y
PMMA-170-quick	160	170	Y	
PMMA-190-quick	160	190	Y	
PMMA-170-DOPO-Me-quick	160	170	Y	
PMMA-170-DOPO-HM-quick	160	170	Y	
PMMA-190-DOPO-HM-quick	160	190	Y	

**Table 2 polymers-17-01254-t002:** Preparation formula of flame-retardant PPO.

Sample	PPO (g)	DVB (g)	DCP (g)	DOPO-Me (g)	DOPO-HM (g)
PPO	20	1	0.4		
PPO-DOPO-Me					
10%	18	0.9	0.36	2	
20%	16	0.8	0.32	4	
30%	14	0.7	0.28	6	
PPO-DOPO-HM					
10%	18	0.9	0.36		2
20%	16	0.8	0.32		4
30%	14	0.7	0.28		6

**Table 3 polymers-17-01254-t003:** Df values of DOPO-Me and DOPO-HM at 20 GHz.

Sample	Df at 20 GHz			
	DOPO-Me		DOPO-HM	
	crystal	amorphous	crystal	amorphous
from FR	0.000733		0.000533	
from PS/FR		0.0129		0.0185
from PMMA/FR		0.0135		0.0168

**Table 4 polymers-17-01254-t004:** Effects of flame retardant and processing temperature on T_1_, Df and *T_g_*.

Sample	T_1_	Df at 20 GHz	*T_g_* (°C)
PMMA-170-slow	24.6	0.003254	110.7
PMMA-170-quick	27.0	0.004110	113.4
PMMA-190-quick	29.8	0.004287	116.1
PMMA-170-DOPO-Me-quick			
0%	27.0	0.004110	113.4
0.25%	25.1	0.003670	112.3
0.5%	24.5	0.003502	111.9
1%	22.1	0.003644	109.8
5%	28.1	0.004008	101.8
10%	28.4	0.004688	93.7
20%	/	0.005223	76.1
PMMA-170-DOPO-HM-quick			
0%	27.0	0.004110	113.4
0.1%	/	0.004097	115.0
0.25%	29.6	0.003840	114.0
0.5%	30.2	0.004321	111.9
1%	29.8	0.004140	110.8
5%	/	0.004761	103.7
10%	28.9	0.005580	92.6
20%	/	0.006626	84.6
PMMA-190-DOPO-HM-quick			
0%	29.8	0.004287	116.1
0.25%	27.6	0.003175	113.6
0.5%	27.1	0.003082	113.0
1%	30.3	0.002638	112.8
5%	29.1	0.004026	102.6
10%	/	0.005801	93.0
20%	/	0.006577	84.7

## Data Availability

The original contributions presented in this study are included in the article. Further inquiries can be directed to the corresponding author.

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
