# Peer review of "Effects of Organophosphorus Flame Retardants on the Dissipation Factor of Flame-Retardant Polymers"

_polymers, 2025, doi:10.3390/polym17091254_

Round 1
Reviewer 1 Report
Comments and Suggestions for Authors
In electronic devices, a range of different materials are used in printed circuit boards as a base material. One important property of these materials is their flame retardancy. In order to deliver the required degree of flame retardancy, the base materials are often doped with large quantities of phosphorous -containing compounds such as DOPO. However, this has an (understudied) impact on the base material’s dielectric properties and has significant cost implications. In this research, the authors look to clarify the impact that the incorporation of a hydroxyl group as part of the flame retardant has on dielectric losses (something that is not easy to predict).
To achieve this, the authors explore the impact on dielectric losses of hydroxyl incorporation into flame retardants and flame retardant-blended polymers. 3 different polymers and a large number of different flame retardant-blended polymers (Tables 1 and 2) are studied. After preparation 1H and 31P NMR, melting point, crystallinity, glass temperature transition, density of flame retardants, dielectric properties and solid state NMRs were used to test the novel materials.
In brief, comparing the flame reatrdants alone, the hydroxymethylated DOPO lowered the dielectric losses (Df) compared to the methylated DOPO. This is rationalised based on intermolecular hydrogen bonding between P=O and OH groups leading to the slowing down of dipole reorientation in an alternating electric field. Surprisingly hydroxymethylated-DOPO also lead to a lower Dk value compared to methylated DOPO.
The comparison was then repeated with the flame retardants blended into polystyrene (used as a model for other polymers that are commonly used). Across a series of detailed studies it is concluded that in non-polar polymers hydroxymethylated DOPO leads to lower dielectric losses. Figure 3 presents this data but it is a bit hard to follow. An insert of the low percentage (0-1%) region would really help the reader follow the story (and see what the authors are referring to). The figure legend for Table 3 could be improved to clarify which of the loading percentages the data corresponds to.
Analogous studies are then reported using the flame retardants and PMMA. In this section an initial surprising drop in Df as a function of increasing percentage of the blended material is seen for both hydroxymethylated and methylated DOPO. An attempt is made to rationalise this but Figure 5 is a bit confusing (the nomenclature used t, g+ and g- is different from in the text tt, tg, gg). Overall different reasons are used to rationalise the observed effects.
Line 268 solid state NMR
The results section finishes with a discussion of the impact of the flame retardants in PPO.
Overall this is a detailed and interesting study that provides a significantly increased level of understanding of the effect of blending different flame retardants into different polymers on Df. It is logical, well done and important. I therefore recommend publication in this journal with some minor changes suggested in the text above
Author Response
1.Summary
Thank you very much for taking the time to review this manuscript and thank you for your valuable comments. Please find the detailed responses below.
- Point-by-point response to Comments and Suggestions for Authors
In electronic devices, a range of different materials are used in printed circuit boards as a base material. One important property of these materials is their flame retardancy. In order to deliver the required degree of flame retardancy, the base materials are often doped with large quantities of phosphorous -containing compounds such as DOPO. However, this has an (understudied) impact on the base material’s dielectric properties and has significant cost implications. In this research, the authors look to clarify the impact that the incorporation of a hydroxyl group as part of the flame retardant has on dielectric losses (something that is not easy to predict).
To achieve this, the authors explore the impact on dielectric losses of hydroxyl incorporation into flame retardants and flame retardant-blended polymers. 3 different polymers and a large number of different flame retardant-blended polymers (Tables 1 and 2) are studied. After preparation 1H and 31P NMR, melting point, crystallinity, glass temperature transition, density of flame retardants, dielectric properties and solid state NMRs were used to test the novel materials.
In brief, comparing the flame reatrdants alone, the hydroxymethylated DOPO lowered the dielectric losses (Df) compared to the methylated DOPO. This is rationalised based on intermolecular hydrogen bonding between P=O and OH groups leading to the slowing down of dipole reorientation in an alternating electric field. Surprisingly hydroxymethylated-DOPO also lead to a lower Dk value compared to methylated DOPO.
The comparison was then repeated with the flame retardants blended into polystyrene (used as a model for other polymers that are commonly used). Across a series of detailed studies it is concluded that in non-polar polymers hydroxymethylated DOPO leads to lower dielectric losses. Figure 3 presents this data but it is a bit hard to follow. An insert of the low percentage (0-1%) region would really help the reader follow the story (and see what the authors are referring to). The figure legend for Table 3 could be improved to clarify which of the loading percentages the data corresponds to.
Response 1: Thank you for your pointing this out.
We have taken your recommendations and added an inset to better visualize curves at 0-1% loadings.
Page 6, Line 174, Figure 3 has been updated with an inset.
Analogous studies are then reported using the flame retardants and PMMA. In this section an initial surprising drop in Df as a function of increasing percentage of the blended material is seen for both hydroxymethylated and methylated DOPO. An attempt is made to rationalise this but Figure 5 is a bit confusing (the nomenclature used t, g+ and g- is different from in the text tt, tg, gg). Overall different reasons are used to rationalise the observed effects.
Response 2: Thank you for your pointing this out.
The nomenclature of t, g+, g- is used for single bond conformation (which reflects the relationship between main chain C-C bonds connecting to the same structural unit) while tt, tg and gg indicate diad conformations.
Line 268 solid state NMR
Response 3: Thank you for your pointing this out.
We have taken your recommendation and made a necessary correction.
Page 8, Paragraph 4, Line 267. solid NMR => “solid state NMR”.
The results section finishes with a discussion of the impact of the flame retardants in PPO.
Overall this is a detailed and interesting study that provides a significantly increased level of understanding of the effect of blending different flame retardants into different polymers on Df. It is logical, well done and important. I therefore recommend publication in this journal with some minor changes suggested in the text above
Reviewer 2 Report
Comments and Suggestions for Authors
To improve the performance of modern electronic devices, materials with low dielectric properties are crucially important. In addition, these materials are also required to have flame retardancy to ensure device safety.
In this manuscript, the authors investigate the relationship between the structures of flame-retardant (FR) additives and the dielectric dissipation factors (Df values) of FR/polymer blends, with a particular focus on the presence or absence of an OH group in FRs. Generally, the OH group would deteriorate the dielectric properties due to its high polarity. However, the authors have found that the OH groups can reduce the Df values in specific cases, which is a particularly interesting finding of this study. According to the authors’ discussion, hydrogen bonds of the OH groups would suppress the mobility of polar groups, thereby improving the dielectric properties.
Considering the recent strong demand for materials with both low dielectric properties and flame retardancy, as well as the unusual relationship between chemical structures and dielectric properties described in this manuscript, the reviewer believes that this manuscript is worthy of publication in Polymers.
The following comments may help to improve the manuscript:
- In the PS/FR blends, the authors evaluated the amorphous state or crystallinity of the FR using DSC. The reviewer recommends to add the DSC thermograms into the main text. These data may be provided in the Supporting Information; however, the reviewer could not find the Supporting Information itself.
- Indeed, the proposed mechanism that the interaction of the OH groups suppresses the mobility of the polar groups, thereby decreases the Df values, would be reasonable. To support this speculation, some spectroscopic evidence should be provided. For example, IR spectroscopy could reveal whether the OH groups form hydrogen bonds in the FR/polymer blends.
- The flame retardant performance (e.g., UL-94 ratings) of the materials should be evaluated, and the reviewer recommends PPO-DOPO-HM as a sample. The results would attract the reader’s interests because the end-group modified PPO is practically used for PCB materials.
- Figures 6 and 7 should be switched each other because the Df of PPO/FR (shown in Figure 7 in the original manuscript) is mentioned before the discussion of the predicted Df values of PPO/FR at 20 GHz (shown in Figure 6 in the original manuscript).
Author Response
1.Summary
Thank you very much for taking the time to review this manuscript and thank you for your valuable comments. Please find the detailed responses below.
2. Point-by-point response to Comments and Suggestions for Authors
- In the PS/FR blends, the authors evaluated the amorphous state or crystallinity of the FR using DSC. The reviewer recommends to add the DSC thermograms into the main text. These data may be provided in the Supporting Information; however, the reviewer could not find the Supporting Information itself.
Response 1: Thank you for your pointing this out.
For PS/DOPO-Me, all of them were amorphous and for PS/DOPO-HM, we have given values of crystallinity in our paper (to see Page 6, Paragraph 2-3, lines 183, 199-201). This should not cause confusion. However, for all PS/FR, we have provided their glass transition temperatures (Tg, to see Figure 3). Tg is likely more important since it not only implies qualitative crystallization of PS/FR (the lower the value of Tg is, the lower crystallinity of the FR in blends due to plasticizing actions), but also allows us to deduce the interaction between PS and FR.
2. Indeed, the proposed mechanism that the interaction of the OH groups suppresses the mobility of the polar groups, thereby decreases the Df values, would be reasonable. To support this speculation, some spectroscopic evidence should be provided. For example, IR spectroscopy could reveal whether the OH groups form hydrogen bonds in the FR/polymer blends.
Response 2: Thank you for your pointing this out.
We have actually measured IR spectroscopy of flame retardant polymers. However, just as pointed out in literatures (for example, Mater Sci: Mater Electron (2024) 35:1419), IR absorption of C=O in PMMA is insensitive to the hydrogen bonding in the solid state. No significant changes in peaks positions were observed. Instead, we used solid NMR to probe changes of relaxation time of C in C=O, which is a more suitable method to probe hydrogen bondings. (For using solid NMR to investigate hydrogen bonded complexes, please see Macromolecules 2012, 45, 6015−6026).
3. The flame retardant performance (e.g., UL-94 ratings) of the materials should be evaluated, and the reviewer recommends PPO-DOPO-HM as a sample. The results would attract the reader’s interests because the end-group modified PPO is practically used for PCB materials.
Response 3: Thank you for your pointing this out.
DOPO derivatives have been used as flame retardants for thermosetting PPO (for example, Reference 14-16 in our paper), and DOPO-Me mentioned in our paper has been proposed as a flame retardant for polyurethane (Polym. Adv. Technol., 2011, 22, 5-13.). Thus we assumed that DOPO derivatives can work as flame retardants. However, we understand that many factors can affect the fire performance of DOPO derivatives and that the choice of a right flame retardant depends on its flame retardancy as well as its effect on the physical properties. We do not expect that DOPO-Me or DOPO-HM can be used as a flame retardant in thermosetting PPO because of their high volatility or low thermal stability. Nevertheless, they can be good model compounds for other DOPO derivatives and hence provide a good opportunity for us to investigate the effect of hydroxyl group of DOPO derivatives on dielectric properties, which was our primary point of focus.
4. Figures 6 and 7 should be switched each other because the Df of PPO/FR (shown in Figure 7 in the original manuscript) is mentioned before the discussion of the predicted Df values of PPO/FR at 20 GHz (shown in Figure 6 in the original manuscript).
Response 4: Thank you for your pointing this out.
We have taken your recommendations and made necessary corrections to switch Figures 6 and 7.
Page 14, 1st paragraph, Line 339, Figure 7 to “Figure 6”
Page 14, 2nd paragraph, Line 351, Figure 6 to “Figure 7”
Page 14, corrected Figure 6 has been positioned above corrected Figure 7.

Reviewer 3 Report
Comments and Suggestions for Authors
The manuscript titled "Effects of Organophosphorus Flame Retardants on the Dissipation Factor of Flame Retardant Polymers" needs the following revisions:
- Abstract has to be included with some numerical values from the obtained results.
- The relationship between the PCB design and the use of organophosphorous compounds as flame retardants in polymers is unclear. Kindly give more emphasis in stating the correlation.
- Kindly comment on the glass transition and crystallinity of all configurations.
- Kindly state the rise and fall of the Df of PPO at nearly 9 Hz (Fig. 7)
- Through the conclusion is comprehensive, the limitations and the scope for further research has to be included.
Author Response
1.Summary
Thank you very much for taking the time to review this manuscript and thank you for your valuable comments. Please find the detailed responses below
2. Point-by-point response to Comments and Suggestions for Authors
1. Abstract has to be included with some numerical values from the obtained results.
Response 1: Thank you for your pointing this out.
We did not provide numerical values because our primary aim was to elucidate the effect of an OH group on dielectric properties of flame retardant polymers. To achieve this goal, we have used three typical polymers and investigated the effect of OH group of the flame retardant on them. What we have found is that flame retardants have different effects on the dielectric properties of three polymers. Even for the same polymer, i.e. PMMA, flame retardants firstly lower Df then increase it at high loadings. Thus, the effect is very complicated. We feel that a relationship between structures of flame retardants and dielectric properties of flame retardant polymers is more relevant and more important than numerical values. Therefore, we have outlined the relationship and factors affecting this relationship instead of providing numerical values.
2. The relationship between the PCB design and the use of organophosphorous compounds as flame retardants in polymers is unclear. Kindly give more emphasis in stating the correlation.
Response 2: Thank you for your pointing this out.
In our introduction, we have mentioned that PCB needs to pass the mandatory flammability tests such as a UL-94 standard. This necessitates the use of flame retardants, particularly for combustible materials such as PPO. On the other hand, a trend toward non-halogenated flame retardants favors the use of organophosphorus flame retardants. (to see first two paragraphs under Section Introduction).
During PCB design, designers need to choose what kinds of materials and flame retardants are used to fabricate low-loss (Df) PCBs. If designers know dielectric loss of flame retardants and a relationship between the structure of FR and dielectric properties of flame retardant polymers, it will be undoubtedly greatly helpful for them.
By the way, current knowledge about dielectric loss of organophosphorus flame retardants has been very limited. We have succeeded to directly obtain Df values of two DOPO derivatives. These values can be a good starting point to help designers to consider suitable DOPO derivatives as potential flame retardants for low-loss PCBs.
- Kindly comment on the glass transition and crystallinity of all configurations.
Response 3: Thank you for your pointing this out.
For polystyrene and poly(methyl methacrylate) used in our study, they are both amorphous. For two flame retardants, they are crystal, but they can become amorphous after blended into polymers, depending on their interactions with polymers. In general, if a flame retardant works as a plasticizer, crystallinity of the flame retardant is reduced and the glass transition temperature of flame retardant polymers is lowered.
4. Kindly state the rise and fall of the Df of PPO at nearly 9 Hz (Fig. 7)
Response 4: Thank you for your pointing this out.
It is a resonant frequency of PPO. PS and PMMA have a resonant frequency at 8.6 GHz, too. Since we used methacrylate-capped PPO and divinylbenzene to crosslink this PPO, the residual small peak around 9 GHz is understandable.
5. Through the conclusion is comprehensive, the limitations and the scope for further research has to be included.
Response 5: Thank you for your pointing this out.
What is our work’s limitation is currently beyond our comprehension. Our main contribution is to correct misconception about the negative effect associated with hydroxyl group in the field of flame retardants (for example, to see Reference 14-16 in our paper). But as any other work, further research has always been needed to establish the limitations and the scope.